# communications
## engineering

# Vertical GeSn nanowire MOSFETs for CMOS beyond silicon

Mingshan Liu[1], Yannik Junk[1,2], Yi Han[1], Dong Yang[1,2], Jin Hee Bae [1], Marvin Frauenrath[3,4], Jean-Michel Hartmann[3,4], Zoran Ikonic[5], Florian Bärwolf[6], Andreas Mai[6], Detlev Grützmacher [1], Joachim Knoch[2], Dan Buca[1] & Qing-Tai Zhao [1✉]

The continued downscaling of silicon CMOS technology presents challenges for achieving the required low power consumption. While high mobility channel materials hold promise for improved device performance at low power levels, a material system which enables both high mobility n-FETs and p-FETs, that is compatible with Si technology and can be readily integrated into existing fabrication lines is required. Here, we present high performance, vertical nanowire gate-all-around FETs based on the GeSn-material system grown on Si. While the p-FET transconductance is increased to 850 μS/μm by exploiting the small band gap of GeSn as source yielding high injection velocities, the mobility in n-FETs is increased 2.5-fold compared to a Ge reference device, by using GeSn as channel material. The potential of the material system for a future beyond Si CMOS logic and quantum computing applications is demonstrated via a GeSn inverter and steep switching at cryogenic temperatures, respectively.

[1] Institute of Semiconductor Nanoelectronics, Peter Grünberg Institute 9 (PGI 9) and JARA-Fundamentals of Future Information Technologies, Forschungszentrum Jülich, 52428 Jülich, Germany. [2] Institute of Semiconductor Electronics, RWTH Aachen University, 52056 Aachen, Germany. [3] CEA, LETI, MINATEC Campus, F-38054 Grenoble, France. [4] University of Grenoble Alpes, F-38000 Grenoble, France. [5] Pollard Institute, School of Electronic and Electrical Engineering, University of Leeds, Leeds LS2 9JT, UK. [6] IHP- Innovations for High Performance Microelectronics, Frankfurt (Oder) 15236, Germany. ✉email: q.zhao@fz-juelich.de

The past decades have witnessed an enormous increase in information processing and data transfer fueled by the extraordinary progress of micro- and nanoelectronics devices and circuits. This evolution is accelerated even more currently by the rapid development of the Internet of Things[1], neuromorphic computing and quantum computing which require substantially more energy efficient electronics. However, the remarkably successful down-scaling of conventional complementary metal oxide semiconductor (CMOS) technology on silicon is reaching physical and technological limitations. Moreover, Si CMOS devices working at deep cryogenic temperatures, which are used as control and readout circuits of qubits in a quantum computing system face a big challenge of scaling of the subthreshold swing to reduce the applied voltage. All these facts have spurred research towards alternative solutions. While devices relying on a different working principle (e.g., band-to-band tunneling[2,3], negative capacitance transistors[4]) do not yet offer satisfying properties, a larger performance boost of CMOS devices is expected to be obtained by replacing silicon with new, higher carrier mobility semiconductors in order to fulfill the ultra-low power requirements for both high-performance room temperature and cryogenic applications.

A large range of materials including III–V semiconductors[5] as well as more exotic materials, such as carbon nanotubes[6] and 2D materials[7] have been investigated. Among them, indium-based III–V compounds, such as InAs[8,9], InGaAs[2,10], or InSb[9,11], exhibit very high electron mobility of about $10^5$ cm$^2$/Vs and could thus be options for n-channel MOS field effect transistors (MOSFETs). Antimony-based materials, like GaSb and InSb can provide higher bulk hole mobility than silicon[12], however, the poor high-k/III–V interface with high density of interface states ($D_{it}$) degrades the p-channel MOSFET drastically[13]. On the other hand, Germanium provides the highest hole mobility among bulk semiconductors[14–17]. However, its electron mobility is rather low. In addition, Ge suffers from a reduced maximum possible donor concentration, a high density of interface states at the high-k dielectric/Ge interface and Fermi level pinning at metal/Ge contacts[18–22]. Hence, heterointegration of III–V semiconductors with Ge has been proposed to simultaneously benefit from high-performance n-/p-channel MOSFETs[12,23]. This is a formidable task due to the mostly incompatible processing technologies of the two material classes. As a result, CMOS functionality has not yet been demonstrated so far in a new material system that can be integrated monolithically on silicon and provides high mobilities for both electrons and holes. From this point of view, newly developed group IV GeSn alloys are highly attractive for future nanoelectronics since they exhibit a number of unique properties[24].

GeSn alloys offer a tunable energy bandgap by varying the Sn content and adjustable band off-sets in epitaxial heterostructures with Ge and SiGe. In fact, a recent report has shown that the use of Ge$_{0.92}$Sn$_{0.08}$ as source on top of Ge nanowires (NWs) enhances the p-MOSFET performances[25]. Lowering the band edge of the conduction band Γ-valley yields the advantage of low effective masses and thus high electron mobilities, as demonstrated in planar long channel GeSn n-FETs[26–28] and FinFETs[29,30]. Even more, at about 8 at% Sn composition[31] for a cubic lattice, or 5 at% Sn under 1% biaxial tetragonal tensile strain[32], the GeSn alloy becomes a direct bandgap semiconductor, a unique property in group-IV semiconductors. This property was recently exploited leading to breakthrough results in photonics, like optically and electrically driven GeSn lasers and mid-infrared imagers integrated on Si[32–34]. Furthermore, pioneering works on spin–orbit coupling, spin transport[35], and thermoelectric properties[36] of GeSn underline the potential of such alloys. In addition to their unprecedented electro-optical properties, a major advantage of GeSn binaries is also that they can be grown in the same epitaxy reactors as Si and SiGe alloys, enabling an all-group IV optoelectronic semiconductor platform that can be monolithically integrated on Si. However, despite all these advantages and research interests, CMOS functionality has not been demonstrated yet in GeSn semiconductors.

This work presents top-down fabricated vertical GeSn-based gate-all-around (GAA) nanowire MOSFETs (VFETs) with NW diameters down to 25 nm. Two epitaxial heterostructures, GeSn/Ge/Si and Ge/GeSn/Ge/Si, are designed to facilitate the co-optimization of p- and n-VFETs, respectively. The GeSn- based devices are compared with all-Ge devices with identical fabrication and benchmarked against literature data. Finally, CMOS functionality is demonstrated by a GeSn-based hybrid CMOS inverter. Last but not least, the same GeSn n-VFET devices show exciting switching properties at low temperatures closing the requirements for cryogenic quantum computing. The present advances presented here are an important step to bring the GeSn semiconductor into CMOS electronics and, together with the successful research in GeSn-based photonics, may finally lead to the long-desired entirely group-IV monolithically integrated electronic-photonic circuits.

## Results and discussion

**GeSn/Ge CMOS concept.** The CMOS concept discussed in this work, as shown in Fig. 1a, is based on the use of different heterostructures designed to yield high-performance p- and n-type VFETs (Fig. 1b) considering high mobility channels, highly doped source/drain regions, low contact resistance on top of the nanowires, and reduced gate induced drain leakage. For p-VFETs, the channel can be Ge or GeSn. First, a simple design like p$^+$-Ge$_{0.92}$Sn$_{0.08}$/Ge is adopted for p-VFETs, where Ge that already provides high hole mobility is used as channel, and the source is the smaller bandgap Ge$_{0.92}$Sn$_{0.08}$ alloy in order to improve the carrier injection and reduce the large NW top contact resistance[25]. The drain region is again Ge to reduce the gate-induced drain leakage by band-to-band tunneling which increases exponentially with the bandgap reduction[3]. For n-VFETs, n$^+$-Ge$_{1-x}$Sn$_x$/i-Ge$_{1-y}$Sn$_y$/n$^+$-Ge$_{1-x}$Sn$_x$ ($x \le y$) heterostructures are designed. Here, the Ge$_{1-y}$Sn$_y$ layer is used as the high electron mobility channel, and the relative larger bandgap Ge$_{1-x}$Sn$_x$ layer ($x < y$) forms the source/drain regions to reduce the gate-induced drain leakage. For a systematic comparison Ge as source/drain regions for n-VFETs were firstly fabricated to underline the GeSn channel electron mobility improvements and to better compare with the later discussed all-GeSn-VFETs. There, the GeSn source/drain regions can have the additional advantage of allowing higher n-type doping thus much lower contact resistance in comparison with Ge source/drain.

The vertical MOSFET design enables the exploitation of electronic band engineering and in situ doping via epitaxial growth with defect-free source/channel/drain interfaces[25]. Such GeSn/Ge and Ge$_{1-x}$Sn$_x$/Ge$_{1-y}$Sn$_y$/Ge$_{1-x}$Sn$_x$ heterostructures can be realized by selective epitaxy, a well-developed process option for Si-based materials. Here, in order to demonstrate the concept easily, the p-VFETs and n-VFETs were fabricated separately on dedicated grown wafers.

The epitaxial stacks are grown by reduced pressure chemical vapor deposition (RP-CVD) method on Ge-buffered 200 mm Si (100) wafers. Details on layers' growth and their characterization are given in Supplementary Note 1 and Figs. S1 and S2. Patterning of GeSn/Ge stacks into thin vertical NWs results in anisotropic strain relaxation, leading to changes in electronic band energies. The lattice strain of as-grown structures was extracted from X-ray diffraction while the tetragonal in-plane and out-of-plane strains in the NW are modeled using finite-

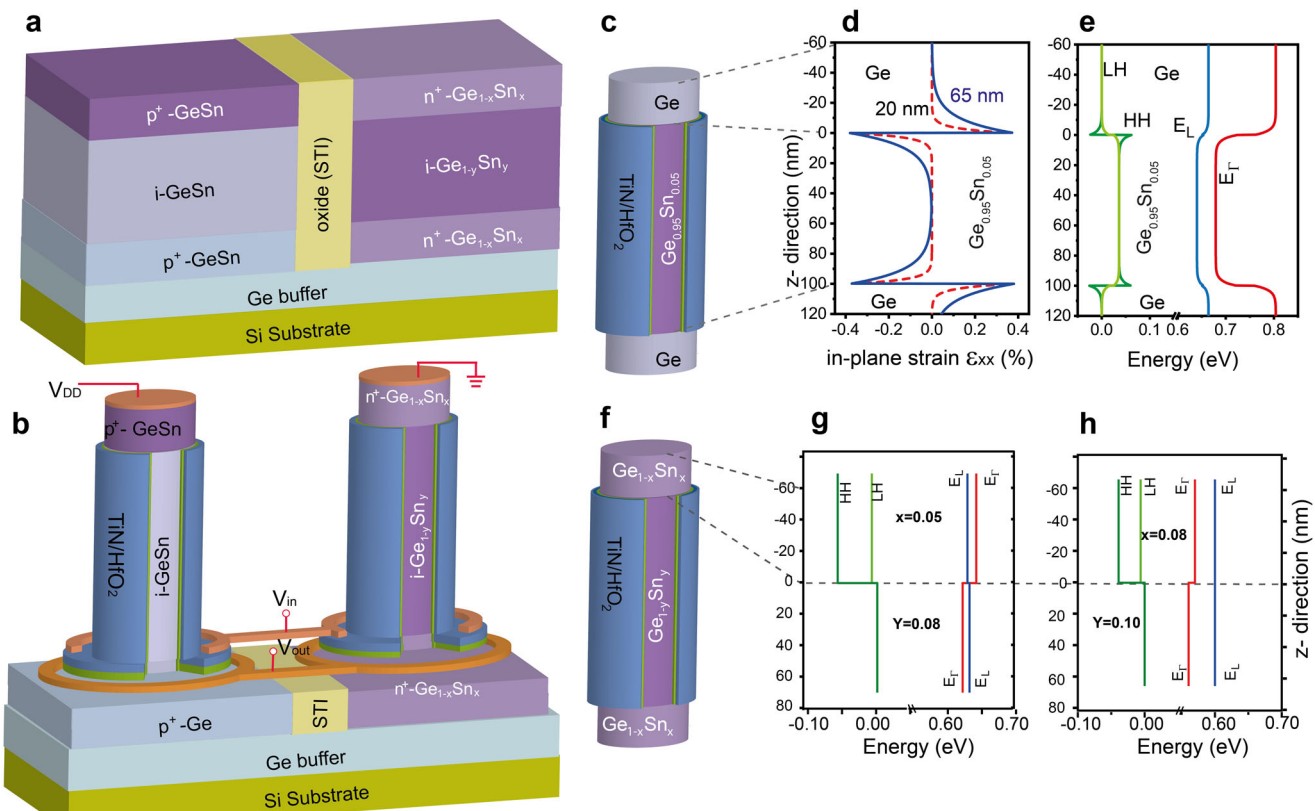

**Fig. 1 Description of the GeSn CMOS concept. a** Schematic cross sectional view of GeSn/Ge stacks grown on Si substrates for CMOS processing. **b** Vertical GeSn/Ge Gate-all-around (GAA) nanowire (NW) CMOS inverter based on stacks in **a**. **c** Schematic view of an n-type vertical intrinsic Ge/$Ge_{0.95}Sn_{0.05}$/Ge NW. **d** In-plane strain, $\varepsilon_{xx}$, variation along the Ge/$Ge_{0.95}Sn_{0.05}$/Ge NW z-direction for NWs with 20 nm (red dashed line) and 65 nm (blue line) diameters. **e** Calculated band energies along z-axis for a 20 nm diameter Ge/$Ge_{0.95}Sn_{0.05}$/Ge NW heterostructure. **f** Schematic view of an n-type vertical intrinsic $Ge_{1-x}Sn_x$/$Ge_{1-y}Sn_y$/$Ge_{1-x}Sn_x$ ($x < y$) NW. **g** Calculated band energies along z-axis for a 20 nm diameter $Ge_{0.95}Sn_{0.05}$/$Ge_{0.92}Sn_{0.08}$ NW heterostructure. **h** Calculated band energies along z-axis for a 20 nm diameter $Ge_{0.92}Sn_{0.08}$/$Ge_{0.90}Sn_{0.10}$ NW heterostructure. Blue: L valley energy $E_L$; Red: $\Gamma$-valley energy $E_\Gamma$; Dark green: energy for heavy holes (HH); Light green: energy for light holes (LH).

element[37] and atomistic modeling[38]. The strain values are then used to calculate the corresponding electronic bands alignment by 8-band k·p method[25,39]. To simplify the computation, no doping is considered in the NW structures. The calculation results are shown in Supplementary Note 2, Tables S1–S3, and Fig. S3.

While the concept of p-VFET was demonstrated in a previous paper[25], here we underline the n-VFETs and the CMOS inverter proof of principle. The in-plane strain, $\varepsilon_{xx} = \varepsilon_{yy}$ along the z-direction and band energy for $Ge/Ge_{0.95}Sn_{0.05}$/Ge vertical NWs ($x = 0.0$, $y = 0.05$) with diameters of 20 nm and 65 nm (Fig. 1c) are shown in Fig. 1d, e. While the L-valley energy $E_L$ is lower than the $\Gamma$-valley energy $E_\Gamma$, the $Ge_{0.95}Sn_{0.05}$ NW channel exhibits an indirect bandgap independently of strain relaxation. Energies for heavy holes (HH) and light holes (LH) have only a slight difference at the interface. For such heterostructure, the in-plane strain at the interface is maximum at the center of the NW and decreases along the radius, reaching zero at the NW surface. The carrier transport along the NW surface is thus different from that in the center, especially for NWs with larger diameters. No other induced strain, e.g., arising from the gate stack itself, is considered in the band structure calculation.

Similar band energy calculations for intrinsic $Ge_{1-x}Sn_x$/$Ge_{1-y}Sn_y$ vertical NWs (Fig. 1f) with $x = 0.05/y = 0.08$ and $x = 0.08/y = 0.10$, and an NW diameter of 20 nm are presented in Fig. 1g, h. The $Ge_{0.95}Sn_{0.05}$ source still exhibits an indirect bandgap while the fully relaxed $Ge_{0.92}Sn_{0.08}$ channel has a direct bandgap of 0.62 eV ($E_\Gamma$) (Fig. 1g). For the $Ge_{0.92}Sn_{0.08}$/$Ge_{0.90}Sn_{0.10}$ NW stack both layers are

direct bandgap semiconductors, and the lower bandgap of 0.56 eV $Ge_{0.90}Sn_{0.10}$ channel provides higher electron mobility.

**Vertical GeSn/Ge GAA NW CMOS process technology.** For n-VFET we start with Ge/$Ge_{0.95}Sn_{0.05}$/Ge ($x = 0/y = 0.05$) heterostructure as indicated in Fig. 1c. A cross-section transmission electron micrograph (TEM) of the Ge/GeSn/Ge heterostructure used for n-VFETs is shown in Fig. 2a. The Ge layer is phosphorous (P) doped while the GeSn channel layer is intrinsic. Vertical GAA NW transistors were processed using a top-down approach employing standard Si CMOS technology (see Methods). The same processing steps and gate stacks are used for the fabrication of both n- and p-type VFETs. Scanning electron microscopy (SEM) images of etched NWs with a height of about 210 nm and diameters of 25 nm and 65 nm are shown in Fig. 2b, c. A cross-section transmission electron micrograph of a final 80 nm diameter Ge/GeSn/Ge GAA vertical NW n-FET with a wrapped-around TiN/HfO$_2$ gate stack is shown in Fig. 2d. An energy dispersive X-ray spectroscopy (EDX) mapping for Ni, Ti and Sn elements is shown in the inset. The top Al/Ti contact is isolated from the TiN gate by a planarization spin-on-glass (SOG) layer. The gate oxide consists of a ~1 nm Al$_2$O$_3$ interfacial layer and with 5 nm HfO$_2$ (inset). The smoothness of the interfaces with inter-diffusion is seen in the high-resolution (HR) transmission electron micrograph shown in the lower inset in Fig. 2e. An energy-dispersive X-ray spectroscopy mapping of elements and a high-resolution transmission electron micrograph image of the top NiGeSn metal contact are provided in

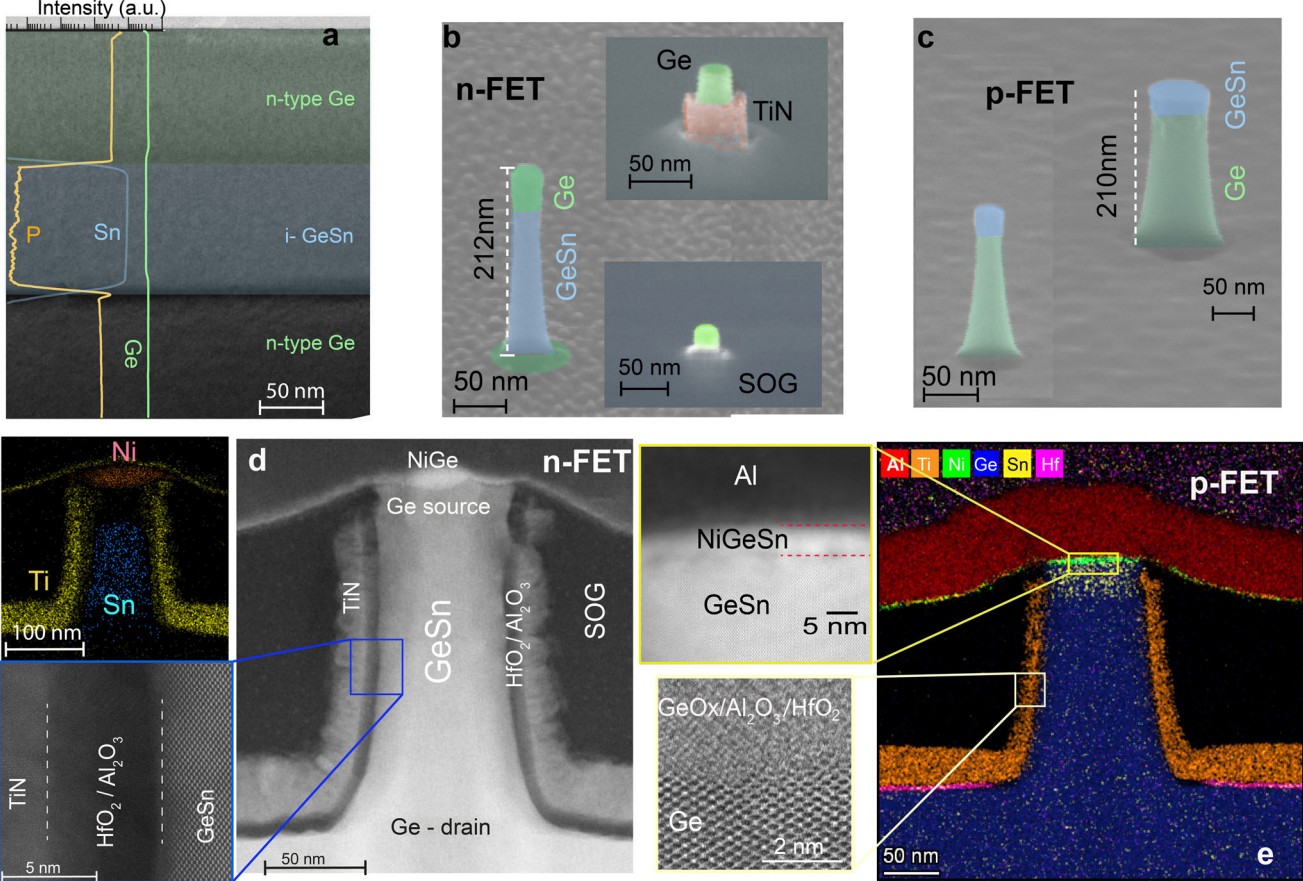

**Fig. 2 Nanowire processing and physical characterization. a** Cross-sectional transmission electron microscopy (TEM) micrograph of the Ge/GeSn/Ge heterostructure used for fabrication of vertical n-FETs (n-VFETs), overlapped with secondary ion mass spectrometry (SIMS) depth profiles of Sn, Ge and P. **b** 3-D scanning electron microscopy (SEM) image of a top-down n-type nanowire (NW). In the upper inset, the spin-on-glass (SOG) is etched to evidence the gate stack -Ge source region. The lower inset shows the top of the NW after the second SOG planarization prior to NiGe contact formation. **c** Overlapped SEM images of GeSn/Ge NWs used for p-VFETs. **d** Cross-sectional TEM image of a vertical Ge$_{0.95}$Sn$_{0.05}$/Ge GAA NW n-VFET. Insets of **d** Energy-dispersive X-ray spectroscopy (EDX) elemental mapping of Ni, Ti, and Sn metals (upper left), High-resolution-TEM (HR-TEM) image showing the sharp interface between GeSn and the GeSnOx/Al$_2$O$_3$/5 nm HfO$_2$ gate stack (bottom left) and the GeSn/Al$_2$O$_3$ interface (bottom right). **e** EDX elemental mapping of a p-VFET with HR-TEM images for the top NiGeSn/GeSn source contact (left top) and for the GeOx/Al$_2$O$_3$/5 nm HfO$_2$ gate oxides on Ge channel.

Fig. 2e for a p-VFET. More process details can be found in Supplementary Note 3 and Fig. S4.

**Electrical characterization of GeSn/Ge GAA NW p-VFET**. A fabricated GeSn/Ge p-VFET is shown schematically in Fig. 3a, where the top GeSn layer is used as source. Different from the previous results[25] where 9 nm thick Al$_2$O$_3$ was used as the gate dielectric here we employed a 5 nm HfO$_2$ and 1 nm Al$_2$O$_3$ as the gate oxide to reduce the equivalent oxide thickness (EOT). The $I_D$–$V_{GS}$ transfer and $I_D$–$V_{DS}$ output characteristics of a single vertical Ge$_{0.92}$Sn$_{0.08}$/Ge GAA nanowire p-VFET with a diameter of 25 nm are shown in Fig. 3b, c. The drain current, $I_D$, is normalized to the NW perimeter. The low subthreshold swing (SS) of 67 mV/decade, the high on-current/off-current ($I_{ON}/I_{OFF}$) ratio and the good saturation reflect the excellent electrostatic control of the gate. The comparison with a p-VFET with a diameter of 65 nm (Fig. 3a) shows the impact of the NW diameter down-scaling: it improves SS but it reduces the on-current, due to a high contact resistance on top of the NW. A peak $G_m$ of >850 μS/μm, much higher than for state-of-the-art GeSn based devices[40,41], is achieved for 65 nm diameter NW p-VFETs (Fig. 3d). The $G_m$ decreases with the decreasing NW diameter, as shown in Fig. 3e, is attributed to contact resistance increase for narrower NW devices. Solutions to further reduce the contact

resistance are using selective growth on top of the nanowire, to increase the contact area for small NWs, and to increase the doping of the GeSn layer. The SS improvement by down-scaling the NW diameter (Fig. 3e) confirms the improved gate controllability for smaller-diameter NWs. Compared to devices from the literature, with Al$_2$O$_3$ as gate oxide[25], the use of a thin and higher-$k$ HfO$_2$ dielectric reduces EOT, and consequently, offers higher on-currents, larger $I_{ON}/I_{OFF}$ ratios and transconductance. A detailed comparison with larger EOT and Ge homojunction NW devices is presented in Supplementary Note 4 and Fig. S5 to further demonstrate the device performance improvements by using GeSn as source and EOT scaling. A SS benchmark for various NW diameters Ge(Sn) NW pFETs is presented in Fig. 3f, showing much better SS than those GeSn devices with a similar NW diameter[40–42]. The present Ge$_{0.92}$Sn$_{0.08}$/Ge NW p-FETs are comparable with in-plane (horizontal) NW GeSn channel p-FETs with 1.5 nm and 3.5 nm diameters[43]. In short, the performance boost of vertical GeSn/Ge NW p-FETs is attributed to the small contact resistance of the GeSn source, 3D nanowire geometry, and excellent surface passivation.

**Vertical Ge/GeSn/Ge GAA NW n-FET characteristics**. The fabrication of Ge n-MOSFETs, as mentioned in the introduction, is very challenging. Here, we show that the use of GeSn channel

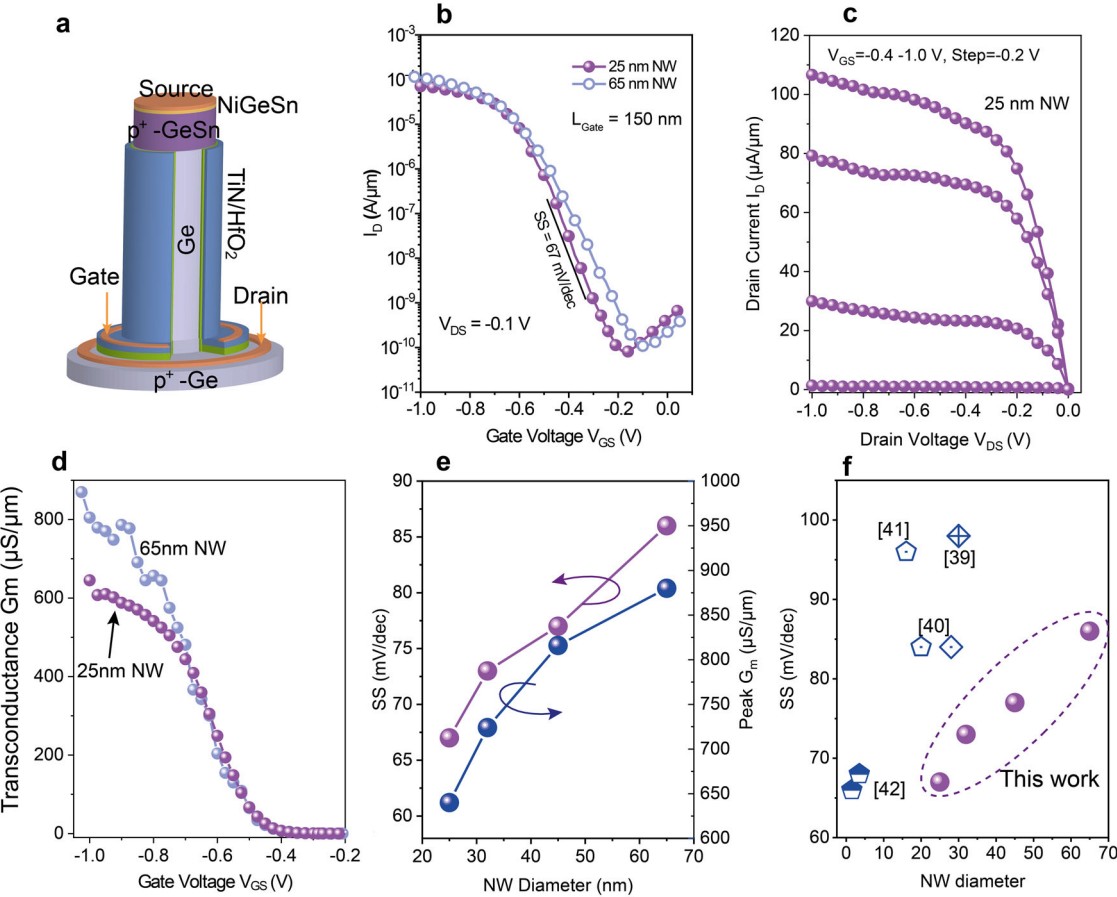

**Fig. 3 Electrical performance of vertical GeSn/Ge Gate-all-around (GAA) nanowire (NW) p-FETs. a** Schematic of a vertical p-FET (p-VFET) with $Ge_{0.92}Sn_{0.08}$ source and Ge channel. **b** $I_D$-$V_{GS}$ transfer characteristics for a p-VFET with NW diameters of 25 nm and 65 nm. **c** $I_D$-$V_{DS}$ output characteristics for a 25 nm NW diameter p-VFET, showing very good saturation. **d** Transconductance $G_m$ of p-VFETs with NW diameters of 65 nm and 25 nm. **e** Subthreshold swing (SS) and peak $G_m$ as a function of NW diameter. The SS improves and the $G_m$ decreases for smaller NW diameters because of the increased top NW contact resistance. **f** Benchmarking of current GeSn p-VFETs with state-of-the-art published GeSn NW p-FETs in terms of the subthreshold swing SS.

considerably improves device performance. The vertical Ge/$Ge_{0.95}Sn_{0.05}$/Ge NW GAA n-type VFET structure is shown in Fig. 4a. The fabrication methodology uses the same processes as for the GeSn/Ge p-VFETs (see Supplementary Note 3 and Fig. S4 for details).

The figures of merit of a vertical $Ge_{0.95}Sn_{0.05}$ channel GAA NW n-VFET, in comparison with a vertical all-Ge homojunction GAA NW n-VFET, are presented in Fig. 4. Both devices have a NW diameter of 25 nm and a gate length of 100 nm. The homojunction Ge device has a SS of 136 mV/dec and an $I_{ON}/I_{OFF}$ ratio of $\sim 1 \times 10^4$ at $V_{DS} = 0.5$ V, which are comparable to those in state-of-the-art horizontal Ge NW n-FETs[22,44]. Using $Ge_{0.95}Sn_{0.05}$ channel improves the SS, which drops down to 92 mV/dec, and results in higher $I_{ON}/I_{OFF}$ ratio ($\sim 1.3 \times 10^4$) and larger on-currents (Fig. 4b). The strong enhancement is clearly reflected also in the transconductance characteristics (Fig. 4c) with $G_{max} \sim 290$ μS/μm peak for GeSn which is 2.5 times larger than the 112 μS/μm obtained for the Ge device. The SS decreases by reducing the NW diameter due to the improved gate control for small NWs (Fig. 4d), while the $G_m$ peak value increases with increasing NW diameter, reaching a high value of 640 μS/μm for 65 nm diameter n-VFET (Fig. 4d), similar to the total resistance as displayed in the Supplementary Information Fig. S6. The higher on-current and transconductance are most likely due to the larger electron mobility in the GeSn channel. An estimation of the mobility ratio between GeSn and Ge channels is given by the Y-function[45]:

$$Y = \frac{I_D}{\sqrt{G_m}} = \sqrt{\frac{W}{L} C_{ox} \mu_0 V_{DS}} \times (V_{GS} - V_{TH}) \quad (1)$$

Where W and L are the gate width (here, the NW perimeter) and channel length, respectively, $C_{ox}$ the gate oxide capacitance and $\mu_0$ the intrinsic mobility. For details about the Y-function see Supplementary Note 5. Therefore, plotting Y as a function of $V_{GS}$ yields a line (Fig. 4d) with a slope A of:

$$A = \sqrt{\frac{W}{L} C_{ox} \mu_0 V_D} \quad (2)$$

The mobility ratio is obtained from the slope of the line, $A_{GeSn}$ for the GeSn device and $A_{Ge}$ for the Ge transistor (Fig. 4e), under the reasonable assumption that the device dimensions W, L, and gate oxide thickness are the same for both devices, in line with the fabrication procedure.

$$\frac{\mu_0(GeSn)}{\mu_0(Ge)} = \left(\frac{A_{GeSn}}{A_{Ge}}\right)^2 = 2.6 \quad (3)$$

It is certainly impressive that a GeSn alloy with just 5 at% Sn improves the electron mobility by 260% compared to the Ge NW device. However, this is in line with the large transconductance improvement. In addition to the higher electron mobility arising from increased electron population of the lower effective mass Γ-

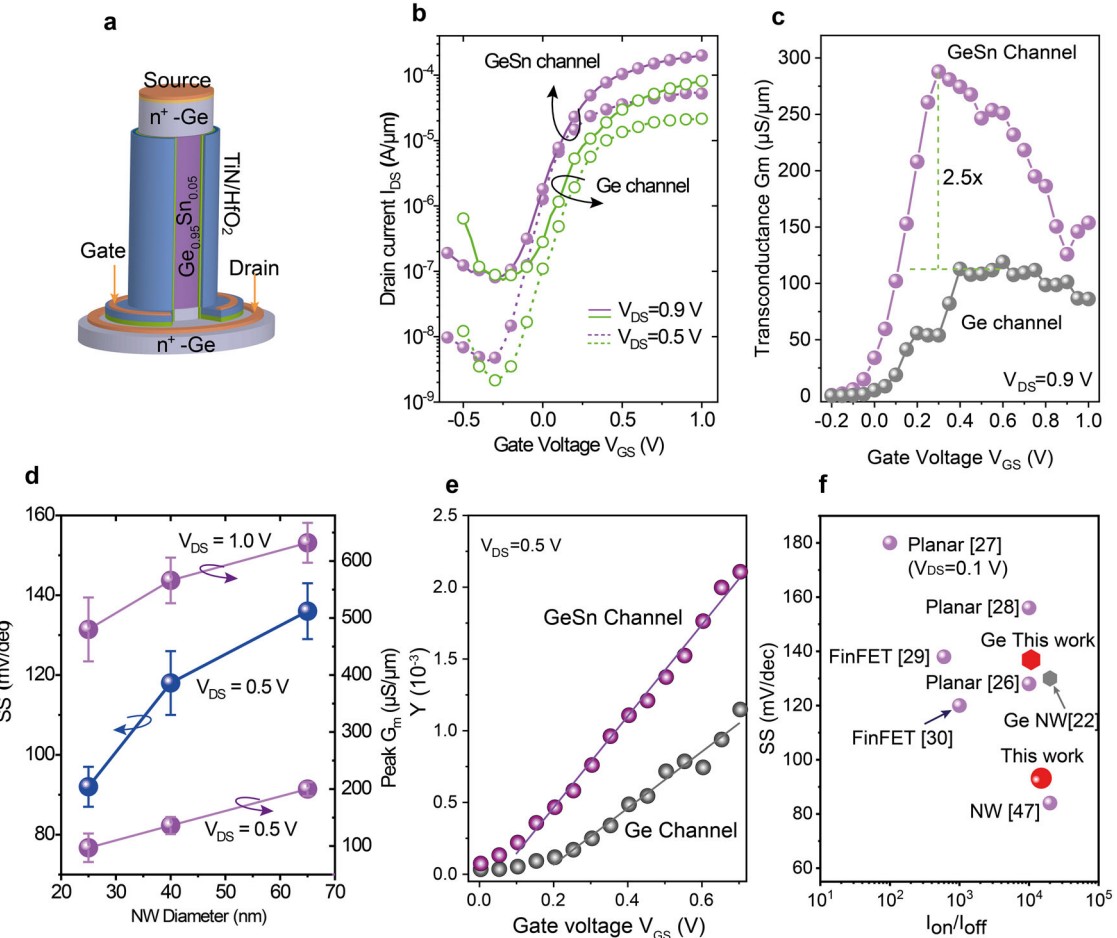

**Fig. 4 Electrical performance of vertical Ge/GeSn/Ge gate-all-around (GAA) nanowire (NW) n-FET. a** Schematic of an n-VFET with a $Ge_{0.95}Sn_{0.05}$ channel and Ge source and drain. **b** $I_D–V_{GS}$ transfer characteristics for a $Ge/Ge_{0.95}Sn_{0.05}/Ge$ n-VFET in comparison with a Ge homojunction n-VFET, both with an NW diameter of 25 nm. **c** 2.5 times higher transconductance, $G_m$, for GeSn channel n-VFET compared to the Ge n-VFET. **d** Subthreshold swing (SS) measured at $V_{DS} = 0.5$ V and peak $G_m$ values as a function of NW diameter, showing improved SS and degraded $G_m$ with decreasing NW diameter. The error bars represent the standard deviations of the measured data from 15 transistors. **e** Plots of the Y-function for the mobility calculation. **f** SS benchmarking of current n-VFETs with state-of-the-art GeSn n-FETs, mostly taken at $V_{DS} = 0.5$ V with the exception of the data in ref. [27] which was given at $V_{DS} = 0.1$ V.

valley, the use of GeSn as channel offers a lower density of interface states with $HfO_2$ dielectrics compared to Ge channel[26,28]. Benchmarking the SS as a function of the $I_{ON}/I_{OFF}$ ratio with state-of-the-art GeSn n-FETs[26–30,46] (Fig. 4f) indicates that the current GeSn n-VFETs are comparable to 17 nm diameter horizontal $Ge_{0.98}Sn_{0.02}$ NW n-FETs from ref. [47] with source/drain doping an order of magnitude higher than here.

**GeSn CMOS inverter**. A CMOS inverter is a basic circuitry of logic integrated circuits (ICs) demonstrating the integration potential of the developed n- and p-VFETs. The GeSn CMOS inverter concept shown in Fig. 1b is experimentally demonstrated using p- and n- VFETs presented above, by externally connecting a GeSn/Ge GAA NW p-VFET and a Ge/GeSn/Ge n-VFET via Al wires, as schematically indicated in the inset of Fig. 5a-3. The $I_D–V_{GS}$ and $I_D–V_{DS}$ characteristics for both n- and p-VFETs are presented in Fig. 5a-1 and a-2. They are symmetric around −0.3 V in terms of $I_{ON}$, SS, and drain-induced barrier lowering. The performance symmetry can be adjusted to 0 V by a proper choice of gate metals with appropriate work functions, as typically done in Si CMOS inverters[47,48]. The voltage transfer characteristics of the inverter for supply voltage, $V_{DD}$, varying from 0.2 V to 1 V, show a very decent transition at around $−0.3$ V $+ V_{DD}/2$.

The shift of −0.3 V is due to the un-matched threshold voltage $V_{TH}$ (cf. Fig. 5b). The apparent degradation in the high $V_{IN}$ regime is caused by the poor saturation of the n-VFET (see $I_D–V_{DS}$ characteristics in Fig. 5a) and high off-currents for the pull-up p-VFET. The voltage gain shows a maximum value of ~18 at $V_{DD} = 0.8$ V (Fig. 5c).

This demonstration of a GeSn CMOS inverter underlines the advantages of GeSn alloys for high-performance nanoelectronics. Further improvements are at hand and include the implementation a self-alignment of gate and channel via an insulating layer between the gate and the substrate, or the use of an all-GeSn heterostructure for the n-type as shown in Fig. 6 and the Supplementary Information Fig. S7 and discussed in the following.

Despite the proof-of-principle for device performance enhancement brought by the GeSn channel, the n-VFET device is still limited by the use of Ge as source and drain (S/D) regions. The high S/D series resistance and super-linear $I_D–V_D$ characteristics at small $V_{DS}$ (Fig. 5a) originate from the low P solubility in Ge. For the case of chemical vapor deposition growth the maximum active P concentration is limited to ~$2 \times 10^{19}$ $cm^{−3}$[49,50], resulting in a Schottky contact and thus, a poor saturation of the $I_D–V_{DS}$ characteristics. This disadvantage is alleviated in GeSn alloys to fabricate all-GeSn n-VFET as illustrated in Fig. 1, where a P doping

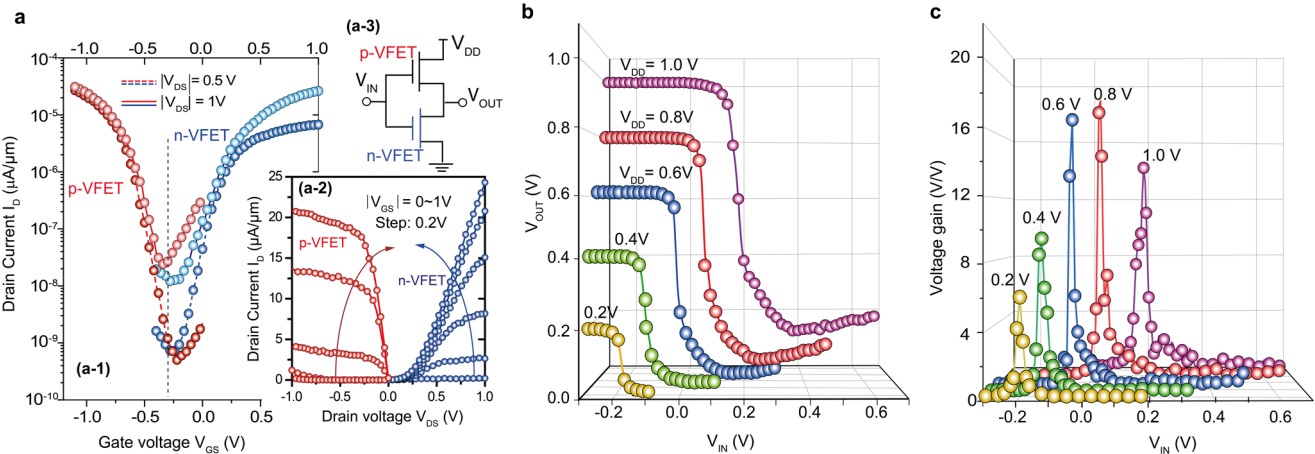

**Fig. 5 GeSn CMOS inverter characteristics. a** Drain current $I_D$ vs gate voltage $V_{GS}$ ($I_D$–$V_{GS}$) (**a-1**) and drain current $I_D$ vs drain voltage $V_{DS}$ ($I_D$–$V_{DS}$) (**a-2**) characteristics of the GeSn/Ge p-VFET and Ge/GeSn/Ge n-VFET forming the CMOS inverter. The p-VFET characteristics are shown in red on the left side of **a-1** and **a-2**, while the n-VFET characteristics are displayed in blue on the right side of the figures. The inset (**a-3**) shows the inverter circuit connections of p- and n-VFETs with the supply voltage $V_{DD}$ applied on the drain of the p-VFET. **b** Voltage transfer characteristics of a hybrid inverter by varying the supply voltage $V_{DD}$ from 0.2 V to 1 V. **c** Voltage gain versus the input voltage $V_{IN}$ of an inverter extracted from Fig. 5b at various supply voltage $V_{DD}$ from 0.2 V to 1.0 V, showing a maximum gain of 18 V/V at $V_{DD} = 0.8$ V.

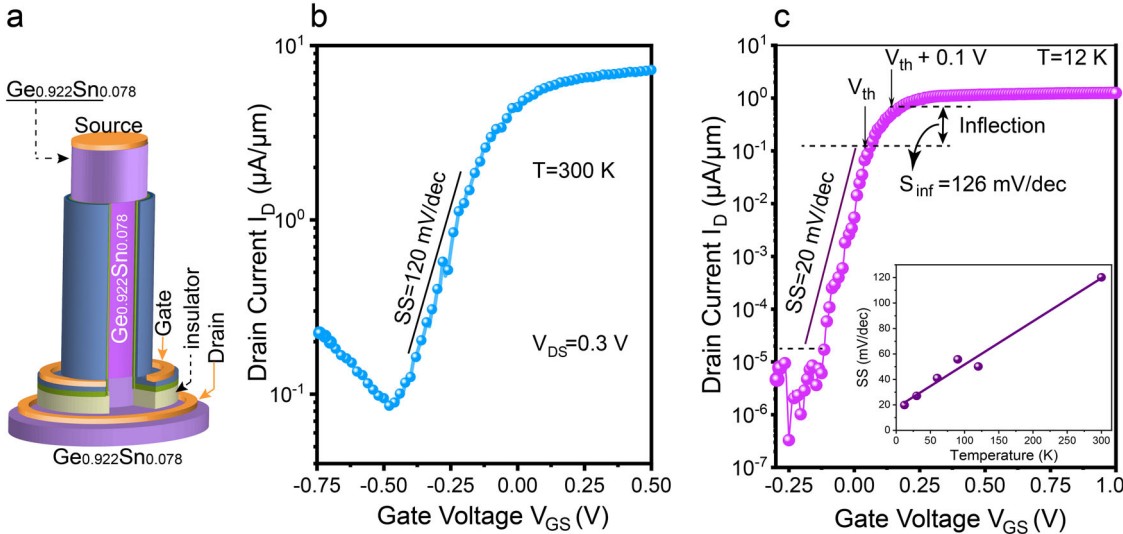

**Fig. 6 Characteristics of an all Ge$_{0.922}$Sn$_{0.078}$ vertical n-FET (n-VFET) with a nanowire (NW) diameter of 50 nm. a** Schematic showing the device structure; **b** $I_D$–$V_{GS}$ transfer characteristics measured at 300 K and, **c** at 12 K. The inset in **c** shows the SS scaling with temperature. The steep slope in the inflection region demonstrates high potential for quantum computing application.

concentration of about $1 \times 10^{20}$ cm$^{-3}$ in GeSn is readily achieved. The use of lower Sn content alloys, i.e., Ge$_{0.95}$Sn$_{0.05}$, as first epitaxial layer relaxes the growth constraints, allowing a thick GeSn layer with a larger Sn content to be pseudomorphically grown. The sketch of an all- Ge$_{0.922}$Sn$_{0.078}$ n-VFET with source/drain layers doped with phosphorous to $7 \times 10^{19}$ cm$^{-3}$ is shown in Fig. 6a. The transfer characteristic of the device as displayed in Fig. 6b, measured at 300 K, shows a subthreshold swing of 120 mV/dec, similar to the Ge/Ge$_{0.95}$Sn$_{0.05}$/Ge n-VFET discussed above. However, the use of a smaller bandgap drain layer increases the gate-induced drain leakage, due to enhanced band-to-band tunneling, leading to a lower $I_{ON}/I_{OFF}$ ratio. This can be solved by using a larger bandgap material, meaning lower Sn content i.e., <5% Sn, while maintaining a high Sn content (>8% Sn) in the channel to improve the electron mobility, as discussed in Fig.1 and Supplementary Information Note 6 and Fig. S7.

Interestingly, the low-temperature measurement shows an additional application direction of the all-GeSn MOSFETs: cryogenic control electronics for quantum computing. Measurements at 12 K not only reduce the off-currents due to the suppressed the trap assisted tunneling and thus, achieve an $I_{ON}/I_{OFF}$ ratio of $10^6$, but offer an SS of 20 mV/dec below the threshold voltage, $V_{th}$, following the Boltzmann scaling of $\frac{kT}{q}$ without saturation (Fig. 6c inset). The inverse slope $S_{inf}$, measured at the inflection region ranging from $V_{th}$ to $V_{th} + 0.1$ V is only 126 mV/dec, much smaller than $S_{inf} = 332$ mV/dec reported[51], in the same voltage range, for Si nanowire MOSFET with a NW cross-section of $20 \times 20$ nm$^2$. This makes the GeSn device very interesting while the conventional cryogenic Si CMOS meets a big challenge called "inflection phenomenon"[52]. In Si CMOS the often-observed saturation of SS in the log$I_D \sim V_{GS}$ linear region at temperatures <50 K and the large $S_{inf}$ necessitate higher applied

drive voltages prohibiting the low power levels needed for cryogenic control electronics (see Supplementary Note 7 and Figs. S8 and S9), the lower $S_{inf}$ and no saturation of $SS$ with the temperature in the all- GeSn n-VFET at cryogenic temperature show high potential for quantum computing applications.

## Conclusions

Vertical gate-all-around GeSn/Ge p-FETs and Ge/GeSn/Ge n-FETs with nanowire diameters down to 25 nm were fabricated and characterized. The small bandgap GeSn alloy used on top of the nanowire considerably boosts the Ge channel p-VFETs performances, offering subthreshold swings as low as 67 mV/dec and very high transconductances of up to 850 µS/µm. For n-VFETs the $Ge_{0.95}Sn_{0.05}$ alloy used as a channel material led to an improved $SS$, a much higher $I_{on}/I_{off}$ ratio, a 2.5 times higher transconductance, and 2.6 times higher electron mobility compared to Ge NW n-VFETs. The symmetry and the high performances of n- and p-VFETs enabled the realization of a GeSn CMOS inverter which showed very good voltage transfer characteristics and high voltage gains. With excellent device performances, high carrier mobilities, band engineering possibilities, steep switching at cryogenic temperatures and Si CMOS compatibility, the GeSn-based CMOS platform provides a path to extend Moore's law beyond the silicon-based era.

## Methods

Ge and GeSn layers were grown by RP-CVD in an industrial cluster tool. Germane, $GeH_4$, was used as precursor gas for pure Ge-epitaxy, and digermane, $Ge_2H_6$, together with tin-tetrachloride, $SnCl_4$, as precursors for GeSn epitaxy. More details about the layer growth can be found elsewhere[31]. The grown layers were examined by Rutherford backscattering spectrometry and X-ray diffraction, in order to obtain information on the Sn-content and the lattice strain, respectively. The high crystalline quality was furthermore confirmed using TEM. Additionally, electrochemical capacitance–voltage measurements were used to obtain the doping concentrations. Further information on material characterization can be found in Supplementary Note 1.

The fabrication of vertical GeSn and Ge nanowire VFETs was performed using standard CMOS processes. After e-beam lithography with Hydrogen Sylses-quioxane as photoresist, reactive ion etching using $Cl_2/Ar$ (4/24 sccm) plasma was performed to form vertical NWs. Digital etching consisting of multiple cycles of self-limiting $O_2$ plasma oxidation at room temperature and diluted HCl stripping was used to shrink the GeSn/Ge and Ge/GeSn/Ge NW diameters and smoothen the NW surfaces. More details can be found elsewhere[25]. Atomic layer deposition was used to wrap $HfO_2/Al_2O_3/Ge(Sn)O_x$ dielectrics around the NWs. The final EOT after post-oxidation process was ~2.4 nm. 40 nm thick TiN deposited by sputtering with argon formed the gate metal. Then, planarization was performed by spin-coated SOG with curing at 350 °C followed by isotropic back-etching in a reactive ion etching chamber with $CHF_3$. The exposed top gate stack was removed by an optimized $Cl_2/SF_6$ etching recipe. Subsequently, a second SOG spin-coating and planarization were performed to isolate the gate stack and top contact. Ni was deposited for top contact formation followed by a forming gas annealing step at 300 °C to form NiGe(Sn) and thereby reduce the top contact resistance. Finally, the device fabrication ended with Ti/Al metallization using lift-off technology after contact window opening and post-metallization annealing.

For characterization, the device current–voltage (I–V) characteristics were measured using a Keithley 4200 Semiconductor Characterization System. For low-temperature measurements, a cryogenic probe station from FormFactor was used, together with an Agilent E5270B for current-voltage characteristics measurements. Liquid He was used for cooling.

## Data availability

The data that support the findings of this study are available from the corresponding author upon reasonable request. Source data underlying the graphs and charts presented in the main figures are included in "Supplementary Data 1".

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

## Acknowledgements

The authors thank the German Federal Ministry of Education (BMBF) for partially supporting the activity via ForMikro - SiGeSn nanoFET project.

## Author contributions

Q.Z. and D.B. planned the device and the experiments. M.F., J.M.H., and D.B. worked on the epitaxy and characterization of the material. M.L. and Y.J. performed the device fabrication and characterization. D.Y. and Y.H carried out part of the device characterization. Z.I. performed the band structure calculation. J.H. B, F.B., and A.M. performed the SEM, TEM, EDX, and SIMS characterization. J.K., D.G., and Q.Z. supervised the work and coordinated device fabrication and data interpretation. D.B., J.K., and Q.Z edited the manuscript and all authors discussed and corrected the manuscript.

## Funding

## Competing interests

The authors declare no competing interests.
