## [Peer Review File · Communications Engineering]

Reviewers' comments:

Reviewer #1 (Remarks to the Author):

The paper reports the potential of GeSn based nanowires as MOSFETs, which is not only interesting, but also very important for next generation electronics. The results obtained are very important and properly benchmarked, so we consider the paper is acceptable based solely on what is written in this paper.

However, I noticed that the content of the section on p-MOSFETs is largely the same as that in ACS Applied Nano Materials (10.1021/acsnm.0c02368) and ECS Transactions (10.1149/10805.0083ecst), which have already been published.

For example, Figs. 1(d) and 1(e) are identical to Figs. 1(g) and 1(h) of ACS paper, respectively, and in Fig. 3, the data from ECS Transactions is used in many places, although the use of colors and the range of graphs have been changed. More to the point, Fig. 2(c) and Fig. 2(e) are different from the SEM and TEM images that appear in ACS paper, in that these are EDX color coded, but do not provide any new information anyway.

Inevitably, the discussions about p-VFET made in this manuscript are almost the same as those reported in previous papers, but they are written as if they are newly reported in this paper. The possibility of being judged as a duplicate publication cannot be denied if this manuscript is published as is.

On the other hand, I think that the section on n-MOSFETs and the CMOS inverter are novel and important results. Therefore, if you can properly cite the published articles correctly to avoid duplicate submissions and clearly separate the previously reported information from the new parts, it would be worthwhile to re-review the article.

Reviewer #2 (Remarks to the Author):

Review summary:

GeSn is a promising channel material beyond Si due to their high carrier mobility and compatibility with Si CMOS technologies. This manuscript reports the demonstration of both vertical GAA p- and n-MOSFETs bead on GeSn material system and shows high gm and low SS. The manuscript also shows an inverter based on GeSn CMOS and cryogenic measurements of GeSn n-FETs. In general, the manuscript is well-structured with interesting results and it is possible to publish in Communications Engineering after addressing the following questions.

1. It is a bit confusing about the sample. Figure 1 shows both n- and p-VFETs are monolithically co-integrated on the same Ge-on-Si wafer, however, the authors also mentioned that an Al wire is used to connect n- and p-FETs for the inverter in page 13. Do you have n-FET and p-FET sample separately and connect them externally or you have both devices on the same sample? The author may need to clarify the sample structures and the processing.

2. In page 2, the authors addressed "...p-channel MOSFETs remain challenging for III-V semiconductors as they lack high hole mobilities." This might not be true since antimony-based III-Vs have high bulk hole mobilities however the technical issues such as bad surface quality (i.e., high Dit) limit the transistor performance. The authors may need to address this better.

3. Several typos in Page 3, "Last but not least, the same GeSn n-VFET devices shown exciting switching properties at low temperatures closing the requirements for cryogenic quantum

computing." Shown  have shown or show? requirements for  requirements for? Computing  computing?

4. In page 4, the authors mentioned that small bandgap GeSn helps reduce the contact resistance of the top electrode in the p-FETs, but why a larger bandgap Ge is used for n-FETs to reduce GIDL? How about the contact resistance in n-FETs and the GIDL in the p-FETs? The authors should probably explain more to make it clear.

5. In page 6, the paragraph below the figure 1 caption is repeated as the previous one.

6. In page 9, "The Gm decrease with the decreasing NW diameter, as shown in Fig. 3e, is attributed to contact resistance increase for narrower NW devices. The SS improvement by down-scaling the NW diameter (Fig. 3e) confirms the improved gate controllability for smaller diameter NWs. " Typo of improvement. Also, here is the trade-off with scaling of the NW diameter. For logic device, steep slope is important while for high-speed devices, high drive current/Gm is important. So how do you balance this with diameter scaling? How do you optimize the device performance by considering this?

7. In page 11 Figure 4f, the author may need to clarify the definition of I_{ON} and I_{OFF} . For the benchmarking, the authors may need to mention the comparison is done at what V_{DS} .

8. In page 14, the author mentioned: "The performance symmetry can be adjusted to 0 V by a proper choice of gate metals with appropriate work-functions, as typically done in Si CMOS inverters." Could you give some references for the Si CMOS inverter with adjusting the work function by comparing gate metals?

9. In page 14, the authors attributed the degradation at high V_{IN} to the poor saturation of n-VFETs from the output characteristics. What are the potential reasons of this poor saturation? Also, from the output characteristics, on-resistance (R_{on}) seems worse in n-FETs as compared to p-FETs, could you also explain the reasons? Do you also have the result of R_{on} as a function of NW diameter?

10. There seems slight ambipolar behavior in both p- and n-VFETs at room temperature but it disappears at 12 K. Does this come from the GIDL? Could you explain the potential cause of this?

Reviewer #3 (Remarks to the Author):

The authors have present high performance, vertical nanowire gate-all-around FETs based on the GeSn-material system grown on Si. It is very interesting work and maybe very helpful to improve the Beyond Moore law. My question is is the p-FET and n-FET has the ability to be integrated in this technology?

Reviewer #1 (Remarks to the Author):

The paper reports the potential of GeSn based nanowires as MOSFETs, which is not only interesting, but also very important for next generation electronics. The results obtained are very important and properly benchmarked, so we consider the paper is acceptable based solely on what is written in this paper.

However, I noticed that the content of the section on p-MOSFETs is *largely the same* as that in ACS Applied Nano Materials (10.1021/acsanm.0c02368) and ECS Transactions (10.1149/10805.0083ecst), which have already been published.

Inevitably, the discussions about p-VFET made in this manuscript are almost the same as those reported in previous papers, but they are written as if they are newly reported in this paper. The possibility of being judged as a duplicate publication cannot be denied if this manuscript is published as is.

On the other hand, I think that the section on n-MOSFETs and the CMOS inverter are novel and important results. Therefore, if you can properly cite the published articles correctly to avoid duplicate submissions and clearly separate the previously reported information from the new parts, it would be worthwhile to re-review the article.

Answer:

Thank you for your straight and fair comment. We agree that the p-VFET “looks” very similar to the previous ACS paper, but we present in this paper an improved p-VFET. In respect to the p-FET in the ACS paper, the major difference in the device structure is: the p-VFET shown here uses scaled and different dielectric: 5nm HfO₂/1nm Al₂O₃ differently deposited as compared to 9nm Al₂O₃ gate dielectrics in the ACS paper. The use of thin and high-k HfO₂ leads to an reduces the EOT to ~2nm, and thus strongly improve the device performance. Moreover, the TEM and EDX images in Fig.2(e) are new and are for the HfO₂ gate dielectrics. These data were never published.

However, we understand that it looked similar and we have now reduced the accent on the p-FET, being just shown to understand the CMOS inverter.

We have also removed the band structure calculation results (Figs.1(d) and 1(e) related with the p-VFET in the previous version). This gives us the opportunity to better underline the n-FET concept extending it to the all-GeSn configurations with high Sn content GeSn layer as the high electron mobility channel, and lower Sn content GeSn layer as the source/drain. So we added in the revised version the band energy calculation results for GeSn (5%Sn)/ GeSn(8%Sn) and GeSn(8%Sn)/GeSn(10%Sn), showing direct bandgap channel with indirect band source/drain (5%Sn) and direct GeSn source/drain (8% Sn source/drain) which is related to the all-GeSn n-VFET showed in the last part of the paper.

Based on the above comments we added:

On page 5:

While the concept of p-VFET was demonstrated in a previous paper²⁵. Here we underline the n-VFETs and the CMOS inverter proof of principle.

On page 9:

Different from the previous results²⁵ where 9 nm thick Al₂O₃ was used as the gate dielectric here we employed a 5nm HfO₂ and 1 nm Al₂O₃ as the gate oxide to reduce the equivalent oxide thickness (EOT).

“Compared to literature devices, with Al₂O₃ as gate oxide²⁴, the use of a thin and higher-*k* HfO₂ dielectric reduces EOT, and consequently, offers higher on-currents, larger I_{ON}/I_{OFF} ratios and transconductance. A detailed comparison with larger EOT and Ge homojunction NW devices is presented in Supplementary Information to further demonstrate the device performance improvements by using GeSn as source and EOT scaling.”

Figure 1 caption and in the text discussions were modified accordingly

Reviewer #2 (Remarks to the Author):

Review summary:

GeSn is a promising channel material beyond Si due to their high carrier mobility and compatibility with Si CMOS technologies. This manuscript reports the demonstration of both vertical GAA p- and n-MOSFETs bead on GeSn material system and shows high gm and low SS. The manuscript also shows an inverter based on GeSn CMOS and cryogenic measurements of GeSn n-FETs. In general, the manuscript is well-structured with interesting results and it is possible to publish in Communications Engineering after addressing the following questions.

Answer:

Thank you. It was a pleasure to read such a laudation comment. The work presented summarized a few layers of design, theoretical calculations, epitaxy efforts and of course fabrication challenges. The results of the cryogenic measurements are surprisingly interesting in terms of ultra low power. We are convinced that this is an application field where GeSn can compete with present materials and technology.

1. It is a bit confusing about the sample. Figure 1 shows both n- and p-VFETs are monolithically co-integrated on the same Ge-on-Si wafer, however, the authors also mentioned that an Al wire is used to connect n- and p-FETs for the inverter in page 13. Do you have n-FET and p-FET sample separately and connect them externally or you have both devices on the same sample? The author may need to clarify the sample structures and the processing.

Answer:

Fig.1 shows only a vision, or better said a concept that we have not realized yet but is possible by selective growth process in semiconductor technology. GeSn selective growth was already reported (see. V. Schlyk et al., "Selective growth of fully relaxed GeSn nano-islands by nano-heteroepitaxy on patterned Si(001)", Appl. Phys. Let. Vol 109 (20), 202102, 2016). We also have initial results on selective epitaxy of GeSn but did not reach the maturity to process such a wafer as yet. In this manuscript we demonstrate the concepts for p-VFET and n-VFET separately on different grown wafers. The inverter was formed by connecting p-VFET and n-VFET from different samples with external Al wire-bonding.

To make it clear we added following in the revised text:

On page 4:

"Here, in order to demonstrate the concept easily, the p-VFETs and n-VFETs were fabricated separately on dedicated grown wafers."

On page 13:

"The GeSn CMOS inverter concept shown in Fig. 1b is experimentally demonstrated using p- and n- VFETs presented above, by externally connecting a GeSn/Ge GAA NW p-VFET and a Ge/GeSn/Ge n-VFET via Al wires,"

2. In page 2, the authors addressed "...p-channel MOSFETs remain challenging for III-V semiconductors as they lack high hole mobilities." This might not be true since antimony-based III-Vs have high bulk hole mobilities however the technical issues such as bad surface quality (i.e., high D_{it}) limit the transistor performance. The authors may need to address this better.

Answer:

Yes, true. We have revised and improved the sentence on page 2 and added a new reference [13]:

"Antimony based materials, like GaSb and InSb can provide higher bulk hole mobility than silicon¹². However, the poor high-k/III-V interface with high density of interface states (D_{it}) degrades the p-channel MOSFET drastically¹³."

3. Several typos in Page 3, "Last but not least, the same GeSn n-VFET devices shown exciting switching properties at low temperatures closing the requirements for cryogenic quantum computing." Shown  have shown or show? requirements for  requirements for? Computing  computing?

Answer:

Very sorry for these mechanical errors. We have, hopefully, corrected all of them.

4. In page 4, the authors mentioned that small bandgap GeSn helps reduce the contact resistance of the top electrode in the p-FETs, but why a larger bandgap Ge is used for n-FETs to reduce GIDL? How about the contact resistance in n-FETs and the GIDL in the p-FETs? The authors should probably explain more to make it clear.

Answer:

Thank you for the comments. We agree that it is a little bit confusing. For p-VFET, GeSn is used only at the source (top of the nanowire) to reduce the contact resistance and to improve the hole injection, thus no contribution to GIDL.

For n-VFETs, our aim is to improve the electron mobility by using GeSn as channel because Ge has lower electron mobility. GeSn with higher Sn content can increase further the electron mobility. Thus we design $n^+-\text{Ge}_{1-x}\text{Sn}_x/i-\text{Ge}_{1-y}\text{Sn}_y/n^+-\text{Ge}_{1-x}\text{Ge}_x$ ($x \leq y$) heterostructures. We started with Ge/GeSn/Ge structure in order to test the electron mobility improvement in comparison with full Ge devices. In this case, we can lower the GIDL but the contact resistance is higher. The contact resistance for n-type Ge is mainly caused by the limit of its low doping concentration. Thus in the last part of the paper we use GeSn with less Sn content as source/drain (all GeSn device), which has the possibility of high doping during CVD. The use of GeSn as the drain can increase the GIDL. We have to make a balance between the GIDL and contact resistance. In the paper we show only the concepts without optimization of the GIDL and contact resistance by changing the Sn content at the source/drain.

Based on your comments we changed completely the first paragraph in the section of "GeSn/Ge CMOS Concept".

The discussion on the GIDL of full GeSn n-VFET can be found on page 16:

"the use of a smaller bandgap drain layer increases the GIDL, due to enhanced band-to-band tunneling, leading to a lower I_{ON}/I_{OFF} ratio. This can be solved by using a larger bandgap material, meaning lower Sn content i.e. <5% Sn, while maintaining a high Sn content (>8% Sn) in the channel to improve the electron mobility, as discussed in Fig.1 and SI."

5. In page 6, the paragraph below the figure 1 caption is repeated as the previous one. Answer:

Thank you. The mistake was caused during editing the format of the paper. It was deleted in the revised version.

6. In page 9, "The Gm decrease with the decreasing NW diameter, as shown in Fig. 3e, is attributed to contact resistance increase for narrower NW devices. The SS improvement by down-scaling the NW diameter (Fig. 3e) confirms the improved gate controllability for smaller diameter NWs." Typo of improvement. Also, here is the trade-off with scaling of the NW diameter. For logic device, steep slope is important while for high-speed devices, high drive current/Gm is important. So how do you balance this with diameter scaling? How do you optimize the device performance by considering this?

Answer:

The typo was corrected. Thank you for pointing it. .

The use of GeSn as high mobility channel is to improve the drive currents. The Gm reduction with scaling of the NW dimension is caused by the increase of the contact resistance on top of the NW. This is not we wanted. So device optimization is still needed, for example using selective growth on top of the NW to increase the contact area. Unfortunately, we cannot do this process in our lab.

To answer your comments we added on Page 9 the following:

“Solutions to further reduce the contact resistance are the using selective growth on top of the nanowire, to increase the contact area for small NWs, and to increase the doping of the GeSn layer.”

7. In page 11 Figure 4f, the author may need to clarify the definition of I_{ON} and I_{OFF} . For the benchmarking, the authors may need to mention the comparison is done at what V_{DS} .

Answer:

These data are mostly taken at $V_{DS}=0.5$ V, except the one shown in Ref.[27] which is taken at $V_{DS}=0.1$ V. We added in the caption of Fig.4f.

8. In page 14, the author mentioned: “The performance symmetry can be adjusted to 0 V by a proper choice of gate metals with appropriate work-functions, as typically done in Si CMOS inverters.” Could you give some references for the Si CMOS inverter with adjusting the work function by comparing gate metals?

Answer:

A CMOS inverter normally performs the transition at $V_{DD}/2$ in order to have a large noise margin. This requires a threshold voltage match between p- and n-FETs. Adjusting the threshold voltage by choosing appropriate work functions for n- and p-FET respectively is a standard technology in Si CMOS. P+ and n+ poly Silicon have been used for p- and n-FETs in the past, respectively. For metal gates we added 2 references in the revised version.

48. Jena, B., Dash, S. & Mishra, G. P. *Improved Switching Speed of a CMOS Inverter Using Work-Function Modulation Engineering*. *IEEE Trans. Electron Devices* **65**, 2422–2429 (2018).

49. Chang, W. T., Li, M. H., Hsu, C. H., Lin, W. C. & Yeh, W. K. *Modifying threshold voltages to n- And p- Type FinFETs by work function metal stacks*. *IEEE Open J. Nanotechnol.* **2**, 72–77 (2021).

9. In page 14, the authors attributed the degradation at high V_{IN} to the poor saturation of n-VFETs from

the output characteristics. What are the potential reasons of this poor saturation? Also, from the output characteristics, on-resistance (R_{on}) seems worse in n-FETs as compared to p-FETs, could you also explain the reasons? Do you also have the result of R_{on} as a function of NW diameter?

Answer:

The poor saturation of the n-VFET is caused by the low doping in Ge. As we discussed on page 11-12, the solubility of phosphorus in Ge is low. Due to the Fermi level pinning metal contacts on n-Ge result in a higher Schottky barrier to the electrons. Thus the Ge/GeSn/Ge n-VFET behaves like a Schottky barrier MOSFET, with a superlinear I_D - V_D characteristics at small V_{DS} and a poor saturation. The Schottky contacts also increase the on-resistance. This is why we are trying to fabricate full GeSn n-VFETs with a possibility of high doping in the GeSn source/drain.

A new $R_{on} \sim NW$ diameter for the n-VFET is added in Figure S6.

10. There seems slight ambipolar behavior in both p- and n-VFETs at room temperature but it disappears at 12 K. Does this come from the GIDL? Could you explain the potential cause of this?

Answer:

You are right that the device shows a slight ambipolar behavior at room temperature. This should be caused by GIDL as the bandgap of GeSn is smaller than Ge. We also believe that trap-assisted tunneling (TAT) at room temperature dominates the GIDL. At low temperature TAT is suppressed and GIDL is also lower.

We added in the revised version on page 16:

“Measurements at 12 K not only reduce the off-currents due to the suppressed the trap assisted tunneling (TAT),...”

Reviewer #3 (Remarks to the Author):

The authors have present high performance, vertical nanowire gate-all-around FETs based on the GeSn-material system grown on Si. It is very interesting work and maybe very helpful to improve the Beyond Moore law. My question is is the p-FET and n-FET has the ability to be integrated in this technology?

Answer:

Thank you for your positive comments.

As we shown in Figure 1, p- and n-FETs can be integrated on the same chip using selective growth of GeSn which has already been demonstrated by other groups (for example, V. Schlykow et al., “Selective growth of fully relaxed GeSn nano-islands by nanoheteroepitaxy on patterned Si(001)”, Appl. Phys. Lett.

Vol 109 (20), 202102, 2016). Unfortunately, selective growth of GeSn is not ready in our lab at this moment.

Reviewers' comments:

Reviewer #1 (Remarks to the Author):

I think this manuscript contains very comprehensive study and high academic value. I also feel that the authors responded truthfully to the comments on the first draft.

Let me just make one comment. I think the authors should add detailed explanation about the top-down etching process to form the nanowires.

Reviewer #2 (Remarks to the Author):

I have no further questions or comments and I think it is ready to publish.

Reviewer #3 (Remarks to the Author):

The questions are properly answered. In my opinion, it is ready to be accepted.

Reviewer #1 (Remarks to the Author):

I think this manuscript contains very comprehensive study and high academic value. I also feel that the authors responded truthfully to the comments on the first draft.
Let me just make one comment. I think the authors should add detailed explanation about the top-down etching process to form the nanowires.

Answer:

Thank you for your comments and very positive evaluation to our paper.

We add more nanowire fabrication details in the Section of **Methods** as following:

After e-beam lithography with Hydrogen Silyles Quioxane (HSQ) as photoresist, reactive ion etching using Cl₂/Ar (4/24 sccm) plasma was performed to form vertical NWs. Digital etching consisting of multiple cycles of self-limiting O₂ plasma oxidation and diluted HCl stripping was used to shrink the GeSn/Ge and Ge/GeSn/Ge NW diameters and smoothen the NW surfaces. More details can be found elsewhere ²⁵.

Reviewer #2 (Remarks to the Author):

I have no further questions or comments and I think it is ready to publish.

Answer:

Thank you for your review and very positive evaluation to our paper.

Reviewer #3 (Remarks to the Author):

The questions are properly answered. In my opinions, it is ready to be accepted.

Answer:

Thank you for your review and very positive evaluation to our paper.

REVIEWERS' COMMENTS:

Reviewer #1 (Remarks to the Author):

The question was properly answered and I have no further questions. Thank you.